# Offspring survival changes over generations of captive breeding

Katherine A. Farquharson[1], Carolyn J. Hogg[1] & Catherine E. Grueber [1✉]

Conservation breeding programs such as zoos play a major role in preventing extinction, but their sustainability may be impeded by neutral and adaptive population genetic change. These changes are difficult to detect for a single species or context, and impact global conservation efforts. We analyse pedigree data from 15 vertebrate species – over 30,000 individuals – to examine offspring survival over generations of captive breeding. Even accounting for inbreeding, we find that the impacts of increasing generations in captivity are highly variable across species, with some showing substantial increases or decreases in offspring survival over generations. We find further differences between dam and sire effects in first- versus multi-generational analysis. Crucially, our multispecies analysis reveals that responses to captivity could not be predicted from species' evolutionary (phylogenetic) relationships. Even under best-practice captive management, generational fitness changes that cannot be explained by known processes (such as inbreeding depression), are occurring.

[1] The University of Sydney, School of Life and Environmental Sciences, Faculty of Science, Sydney, NSW, Australia. ✉email: catherine.grueber@sydney.edu.au

Captive breeding is increasingly relied upon to prevent extinction[1,2]. Conservation programs aim to halt evolution so that captive populations remain representative of wild sources[3]. However, even the best efforts cannot fully replicate the wild in captivity. Selective pressures that differ in captivity relative to the wild can drive captive adaptations. Adaptation to captivity may improve population-level fitness in the captive environment if the individuals best suited to captivity are more successful[4]. But, when animals are returned to the wild, captive adaptations may be maladaptive (e.g. selection for tameness), and contribute to the low success of reintroduction programs[5]. Adaptation to captivity has been investigated in multiple fish species and model organisms (for example[6–8]), but is underexplored in conservation settings[9]. Although conservation breeding programs employ strategies to minimise the effects of adaptation to captivity (such as avoiding intentional selection, attempting to replicate natural environments and fragmenting populations)[10], the extent and consequences of adaptation to captivity are largely unknown. Genetic changes as a result of captive breeding have been demonstrated to occur in as little as a single generation in steelhead trout[11], so conservation breeding programs are unlikely to be immune to the effects of adaptation to captivity.

A recent systematic review and meta-analysis investigating birth origin effects on reproductive success revealed captive-born animals have substantially lower reproductive success in captivity than their wild-born counterparts, particularly for offspring survival traits[12]. This result seems to conflict with the expectation of improved fitness in captivity (e.g.[13]); however, many of the studies reviewed considered only the first generation ($G_1$) of captive breeding. The response of species to captive breeding may differ in the first generation relative to later generations, as different pressures may apply. For example, first generation changes may occur as a consequence of non-genetic effects such as husbandry, nutrition or maternal effects[14]. Longer-term, multi-generational changes may instead reflect heritable genetic change. Therefore, it is essential to disentangle first generation and multi-generational changes when investigating adaptation to captivity.

Other genetic factors such as inbreeding can also contribute substantially to fitness changes over time in small populations. Inbreeding depression can affect various life-history traits, such as fertilisation, embryo survival, offspring survival and total lifetime reproductive success[15,16]. In conservation contexts, inbreeding has been demonstrated to reduce offspring survival with no detectable purging to reverse negative fitness effects[17,18]. As a result, strategies to minimise inbreeding are widely applied in conservation contexts[19]. However, few studies of inbreeding also consider generations in captivity: as inbreeding is often positively correlated with time in captivity, other mechanisms of fitness change over time may be confounded with inbreeding[20]. Small sample sizes typical of conservation settings limit statistical power to detect inbreeding depression and to separate confounding factors influencing generational fitness changes.

Globally, the diversity of species bred in captivity must increase if extinctions are to be prevented[21]: while a great variety of species are bred in zoos, this variety does not reflect that of the diversity of threatened species for a range of reasons, which may be challenging to overcome[22]. The species that are currently managed in captivity are not just phylogenetically diverse, but also differ in their life history, reproductive biology, social structure and many other important aspects. Phylogenetic comparative methods can be useful to observe trends and patterns across multiple species, so that the results can be extended to related taxa[23], overcoming the limitations of small sample sizes. Although individual conservation breeding programs may have operated for a short time, or with a small population, these programs also record a wealth of data that can be analysed collectively to generate powerful insights[24]. Captive breeding programs routinely use pedigrees to measure and manage genetic diversity and inbreeding[19]. Studbooks recording births, deaths and parentage information can be analysed to retrospectively investigate aspects of reproduction without the need for experimental manipulation of often threatened species. With the increasing accessibility of standardised studbook data through large curated online databases, such as the Zoological Information Management Software (ZIMS)[25], there are now opportunities to examine important traits such as offspring survival at a much larger scale.

In this study, we use studbook (pedigree) data from 15 diverse and long-running conservation breeding programs (37,484 datapoints; Table 1) to investigate the drivers of change in offspring survival to reproductive maturity. We specifically aim to examine generational trends, by comparing main effects estimated from all 15 species to individual species-level responses, while controlling for inbreeding[26]. We also aim to disentangle first-generation and multi-generational changes in offspring survival. We find that species differ in their response to generations of captive breeding, with offspring survival increasing in some species, and decreasing in others. We find consistent negative effects of offspring inbreeding on survival, and reveal differences in dam and sire responses to first-generation versus multiple generations of captive breeding. Generational fitness changes are occurring in captivity but remain challenging to predict across taxa.

## Results

**Pedigree information provides a rich source of data for retrospective analyses of captive breeding trends.** We collected pedigree information from 15 diverse vertebrate captive breeding programs, totalling 58,611 individuals. The earliest record was dated 1850, from the cheetah (*Acinonyx jubatus*) studbook (Table 1). All studbooks represent captive breeding programs that are currently managed for demographic and genetic soundness by regional and/or global zoo associations. From pedigree records, we extracted metrics related to individual inbreeding, age and the number of generations of captive breeding (Table 1). The number of generations in captivity refers to the captive generations experienced by an individual, where wild-born animals are assigned $G_0$ and animals in descendant generations are given the average generation of their parents plus 1. Generations varied across our dataset, as expected due to variation in the age of the programs, life-history of the species (fast breeding species will experience more generations over the same time period than slow breeding species), and availability of wild animals (which reduces generations in captivity). For example, the radiated tortoise (*Astrochelys radiata*) studbook had an average sire generation in captivity of 0.135 (SD = 0.342), compared to the average dam generation of 4.218 (SD = 2.758) in the European mink studbook (*Mustela lutreola*) (Table 1).

We used two main approaches to examine drivers of offspring survival in captivity:

1. Main (overall) effect: generalised linear mixed models (GLMMs) were used to quantify the main effects of seven parameters—sire, dam, and offspring inbreeding; sire and dam age; and sire and dam generations in captivity, on the response variable (offspring survival)—across the dataset as a whole (i.e. model built upon data from all 15 species). A positive regression estimate indicated a positive effect of the predictor on offspring survival, conversely, a negative regression estimate indicated a negative relationship with offspring survival. An estimate with confidence intervals excluding zero was interpreted as a statistically significant

**Table 1 Summary statistics for the fifteen studbooks included in the main analysis ($N = 37{,}484$ individuals).**

| Species (scope of studbook[1]; IUCN status) | $N$[2] | $N_e$[3] | First record (year) | Age (days) at maturity female; male | Pedigree f (mean, sd, max) dam (D); sire (S); offspring (O) | Captive generations (mean, sd, max) dam (D); sire (S) | Age at breeding (days) (min, mean, max, sd) dam (D); sire (S) |
|---|---|---|---|---|---|---|---|
| Prehensile-tailed skink *Corucia zebrata* (AZA; N/A) | 534 | 40.1 | 1969 | 4502[4]; 5702[4] | D: 0, 0, 0<br>S: 0, 0, 0<br>O: 0.012, 0.049, 0.250 | D: 0.208, 0.431, 1.750<br>S: 0.301, 0.527, 1.750 | D: 636, 3696, 9114, 1808<br>S: 576, 3897, 13000, 1979 |
| Western swamp tortoise *Pseudemydura umbrina* (ZAA; CR) | 581 | 28.9 | 1949 | 2910[4]; 2100[4] | D: 0, 0, 0<br>S: 0, 0, 0<br>O: 0, 0, 0 | D: 0.184, 0.388, 1<br>S: 0.201, 0.401, 1 | D: 3261, 11686, 22388, 5062<br>S: 2156, 11652, 25243, 5999 |
| Radiated tortoise *Astrochelys radiata* (AZA; CR) | 757 | 58.0 | 1900 | 3720[4]; 3000[4] | D: 0, 0, 0<br>S: 0, 0, 0<br>O: 0, 0, 0 | D: 0.178, 0.404, 1.5<br>S: 0.135, 0.342, 1 | D: 3907, 12951, 22795, 5018<br>S: 3043, 12716, 23835, 5474 |
| Tasmanian devil *Sarcophilus harrisii* (ZAA; EN) | 1111 | 163.4 | 1982 | 730; 730 | D: 0.012, 0.052, 0.375<br>S: 0.007, 0.033, 0.250<br>O: 0.016, 0.049, 0.375 | D: 1.348, 1.204, 5.408<br>S: 1.084, 1.152, 4.500 | D: 231, 944, 2470, 295<br>S: 231, 1177, 2466, 414 |
| Cheetah *Acinonyx jubatus* (INTL; VU) | 4932 | 222.5 | 1850 | 456; 456 | D: 0.010, 0.034, 0.250<br>S: 0.011, 0.042, 0.250<br>O: 0.022, 0.055, 0.268 | D: 1.567, 1.387, 5.353<br>S: 1.339, 1.290, 5.156 | D: 651, 2313, 6084, 770<br>S: 590, 2535, 6329, 981 |
| Meerkat *Suricata suricatta* (AZA; LC) | 1659 | 17.0 | 1908 | 365; 365 | D: 0.078, 0.132, 0.406<br>S: 0.073, 0.146, 0.434<br>O: 0.115, 0.156, 0.516 | D: 1.828, 1.244, 5.391<br>S: 1.594, 1.348, 5.250 | D: 217, 1793, 4918, 925<br>S: 89, 1836, 4798, 863 |
| Red wolf *Canis rufus* (AZA; CR) | 958 | 108.0 | 1966 | 330[4]; 330[4] | D: 0.036, 0.035, 0.125<br>S: 0.032, 0.034, 0.250<br>O: 0.055, 0.039, 0.250 | D: 2.985, 1.624, 6.059<br>S: 2.935, 1.601, 6.133 | D: 345, 2056, 4033, 818<br>S: 722, 2288, 5499, 1037 |
| African wild dog *Lycaon pictus* (INTL; EN) | 5391 | 81.1 | 1887 | 639; 639 | D: 0.058, 0.117, 0.503<br>S: 0.065, 0.122, 0.503<br>O: 0.107, 0.137, 0.594 | D: 2.211, 1.462, 6.594<br>S: 2.037, 1.477, 6.594 | D: 361, 1698, 6880, 710<br>S: 361, 2070, 6721, 982 |
| Red panda *Ailurus fulgens* (INTL; EN) | 2926 | 382.2 | 1868 | 550; 550 | D: 0.020, 0.035, 0.274<br>S: 0.020, 0.035, 0.298<br>O: 0.036, 0.048, 0.375 | D: 2.828, 1.723, 6.969<br>S: 2.714, 1.767, 7.098 | D: 357, 2005, 5157, 905<br>S: 361, 2329, 6945, 1120 |
| European mink *Mustela lutreola* (EAZA; CR) | 1480 | 74.5 | 1932 | 323; 323 | D: 0.084, 0.111, 0.504<br>S: 0.083, 0.108, 0.481<br>O: 0.117, 0.119, 0.504 | D: 4.218, 2.758, 10.397<br>S: 4.195, 2.955, 10.514 | D: 282, 919, 3309, 567<br>S: 295, 998, 3645, 647 |
| Scimitar-horned oryx *Oryx dammah* (INTL; EW) | 6435 | 216.0 | 1872 | 639; 210[4] | D: 0.086, 0.124, 0.646<br>S: 0.087, 0.126, 0.601<br>O: 0.112, 0.137, 0.675 | D: 2.938, 1.848, 7.870<br>S: 2.962, 1.879, 8.027 | D: 564, 2552, 10429, 1408<br>S: 518, 2478, 8359, 1334 |
| Eastern bongo *Tragelaphus eurycerus isaaci* (INTL; CR) | 2443 | 174.9 | 1931 | 806; 914 | D: 0.066, 0.100, 0.614<br>S: 0.086, 0.106, 0.614<br>O: 0.086, 0.110, 0.614 | D: 2.966, 1.538, 6.812<br>S: 2.933, 1.512, 6.166 | D: 396, 2471, 7004, 1250<br>S: 582, 2607, 7054, 1153 |
| Red-ruffed lemur *Varecia rubra* (INTL; CR) | 1737 | 171.4 | 1959 | 609; 650 | D: 0.085, 0.084, 0.375<br>S: 0.097, 0.086, 0.445<br>O: 0.138, 0.102, 0.445 | D: 2.576, 1.217, 5.484<br>S: 2.651, 1.261, 5.672 | D: 707, 3105, 10925, 1487<br>S: 719, 3414, 10990, 1761 |
| Black-and-white ruffed lemur *Varecia variegata* (INTL; CR) | 3516 | 275.6 | 1959 | 605; 649 | D: 0.046, 0.077, 0.375<br>S: 0.053, 0.081, 0.344<br>O: 0.085, 0.101, 0.438 | D: 2.353, 1.295, 5.969<br>S: 2.175, 1.358, 5.805 | D: 292, 3017, 9911, 1598<br>S: 568, 3270, 14335, 1777 |
| Goeldi's monkey *Callimico goeldii* (INTL; VU) | 3024 | 187.5 | 1913 | 365; 395 | D: 0.019, 0.049, 0.375<br>S: 0.019, 0.049, 0.375<br>O: 0.034, 0.069, 0.500 | D: 2.648, 1.475, 6.326<br>S: 2.621, 1.496, 6.336 | D: 441, 2304, 7763, 1144<br>S: 184, 2471, 7518, 1221 |

[1] INTL: International (WAZA studbook), AZA Association of Zoos & Aquariums, EAZA European Association of Zoos and Aquaria, ZAA Zoo and Aquarium Association Australasia.
[2] IUCN Red List Status[75]: N/A not recorded, EW extinct in the wild, CR critically endangered, EN endangered, VU vulnerable, LC least concern.
[2] With complete data (both parents known, no missing values), 9 outliers removed as per methods.
[3] $N_e$ obtained from PMx using whole studbook prior to data filtering. PMx estimates $N_e$ from Wright[76].
[4] No AnAge record available, so age obtained from PMx.

| Table 2 Model estimates when one offspring is selected from each litter/clutch. | | | |
| --- | --- | --- | --- |
| Predictor | Mean estimate | Mean SE | 95% CI |
| **a) All offspring** | | | |
| Intercept | 0.4610 | 0.1853 | 0.0977, 0.8242 |
| Dam generation | 0.0048 | 0.0203 | −0.0350, 0.0446 |
| Sire generation | 0.0216 | 0.0188 | −0.0152, 0.0584 |
| Dam age at breeding | −0.0624 | 0.0156 | −0.0931, −0.0318 |
| Sire age at breeding | 0.0583 | 0.0159 | 0.0271, 0.0895 |
| Dam $f$ | 0.0020 | 0.0164 | −0.0300, 0.0341 |
| Sire $f$ | 0.0086 | 0.0165 | −0.0237, 0.0409 |
| Offspring $f$ | −0.1632 | 0.0157 | −0.1940, −0.1323 |
| **b) $G_{2+}$** | | | |
| Intercept | 0.4226 | 0.1554 | 0.1181, 0.7272 |
| Dam generation | −0.0322 | 0.0281 | −0.0872, 0.0228 |
| Sire generation | 0.0441 | 0.0278 | −0.0103, 0.0985 |
| Dam age at breeding | −0.0893 | 0.0191 | −0.1267, −0.0518 |
| Sire age at breeding | 0.0787 | 0.0188 | 0.0418, 0.1156 |
| Dam $f$ | −0.0064 | 0.0189 | −0.0435, 0.0307 |
| Sire $f$ | 0.0177 | 0.0189 | −0.0194, 0.0548 |
| Offspring $f$ | −0.1831 | 0.0186 | −0.2196, −0.1465 |

(a) Standardised parameter estimates of pooled ($N = 5$) analyses after randomly sampling one independent offspring from each litter and model averaging ($N = 21,282$ individuals). (b) Standardised parameter estimates of pooled ($N = 5$) independent litter sampling $G_{2+}$ analyses ($N = 16,514$–16,516 individuals, see Methods).

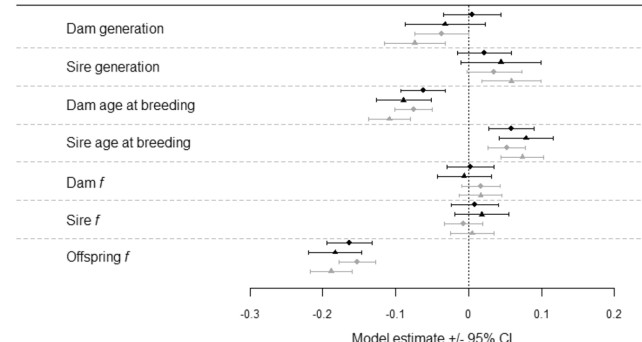

**Fig. 1 Model estimates for offspring survival analyses (±95% CI).** Black circle is the average model estimate of the pooled results ($N = 5$ models) from random selection of one offspring per litter/clutch ($N = 21,282$ individuals), black triangle is the average model estimate of the pooled results ($N = 5$) from $G_{2+}$ subset of this model ($N = 16,514$–16,516 individuals), grey circle is the model estimate of the extended dataset model ($N = 37,484$) after model averaging, grey triangle is the model estimate of the $G_{2+}$ subset of this model ($N = 27,734$) after model averaging. Positive estimates represent a positive relationship with offspring survival to reproductive maturity.

Some species showed strong positive relationships between the number of generations in captivity and offspring survival (e.g. red wolf, African wild dog, western swamp tortoise), whilst others had strong negative relationships (black-and-white ruffed lemur, Tasmanian devil) (Fig. 2b).

**Multi-generational ($G_{2+}$) effects on offspring survival differ between the sexes.** To disentangle fitness changes that may occur in the first-generation of captive breeding from longer-term ongoing fitness changes, we reanalysed our data after removing first-generation offspring (only $G_{2+}$ offspring retained). Most main effects from the overall analysis were similar to those obtained from the model containing all offspring, but with slightly less precision, as expected given the smaller dataset (Table 2b, Fig. 1). However, the main effect for dam generation became negative. When we examined species-level generation effects with the $G_{2+}$ data subset, sire generation reflected a similar pattern to the model with all offspring (Fig. 2b), with high variation between species. However, the species-level random slope estimates for dam generation became slightly negative across all species. Detailed illustrations of species-level random slopes are presented in Supplementary Figs. 4–18.

**Offspring survival is impacted by parental age effects.** Dam age at breeding had an overall negative relationship with offspring survival, while sire age at breeding had a similar-sized positive relationship (Table 2). At the species level, offspring survival was negatively associated with dam age for most species, although the prehensile-tailed skink had a steep positive slope (Fig. 2b). A random slope model could not be fitted to investigate species-level effects of sire age at breeding due to model convergence issues, possibly due to a lack of variation in species responses.

**Discussion**
We used multi-species mixed-effects models to investigate the effect of generations in captivity on offspring survival to reproductive maturity in 15 vertebrate species encompassing over 30,000 data points whilst accounting for inbreeding, parental age, year effects, regions and species phylogeny. Although effects of inbreeding and age were largely consistent with conventional predictions[17,26,27], changes in survival over generations showed

trend at $\alpha = 0.05$. We also quantified phylogenetic signal of the overall survival estimates (to test whether closely related species share a similar response to captive breeding).

2. Species-level effects: random-slope estimates from GLMMs were obtained to quantify variation in the responses of each of the 15 species to captive breeding. As with our main effects, a positive random slope estimate represents a positive relationship with offspring survival for that species. We also examined phylogenetic signal of each of the parameters of interest on offspring survival using the species-level estimates for each parameter.

**Offspring survival changes over generations of captive breeding, independently of substantial inbreeding depression.** Offspring inbreeding ($f$; equivalent to the kinship of the parents) had the strongest negative relationship with offspring survival across the dataset (i.e. main effect; Table 2, Fig. 1). Sire and dam $f$ main effects were both estimated close to zero (Table 2). We found no evidence of phylogenetic signal for survival in captivity, indicating offspring survival could not be predicted by taxonomic relationships (lambda $= 7 \times 10^{-5}$; Fig. 2a). There was no evidence of statistically significant phylogenetic signal for any of the effects of any parameters tested, except for dam $f$ (Supplementary Table 1; Supplementary Fig. 1). Considering species-level effects (random slopes), all species had a negative slope for offspring $f$, with the three primates and the European mink showing the strongest effects of inbreeding (Fig. 2b).

Controlling for inbreeding, dam and sire generation showed no overall effect on offspring survival when considering all 15 species together (Table 2). However, when examining species-level effects, generational effects were highly variable among species.

complex variation among species, between the sexes, and between first and subsequent generations in captivity. Alongside a lack of phylogenetic signal for offspring survival, our analysis implies that intensive human management can drive ongoing change in conservation breeding programs as has been observed in fisheries[6,13], and that these changes will be difficult to predict for any given taxon.

We found a high degree of inter-species variation in the effect of dam and sire generations in captivity on offspring survival from our species-level random-slopes models (Fig. 2b), explaining the absence of any consistent overall effect of generations in captivity in our multi-species model (Fig. 1). These generational changes in survival were observed in multiple species despite controlling for other processes that may result in changes over time such as changes in husbandry[28], population management[29], or the accumulation of inbreeding[30].

It is plausible that generational effects of captivity primarily occur in the first generation, when wild animals are brought into captivity, with little subsequent change. This first-generation change has been observed in some fish species[11,13,31]. Differentiating immediate (but short-lived) change from ongoing, accumulating change, is important for management planning. Previous studies have either not attempted, or been unable, to disentangle first-generation changes from multi-generational change[7,12], although the influence of maternal effects has been accounted for experimentally in model organisms[32]. Our results, comparing our full models to only generation $G_{2+}$ animals (i.e. animals with two captive-born parents) showed that despite the potential for major changes in the first generation, impacts occurred across captive generations in many species.

Changes in offspring survival occurred over multiple generations, but with substantial among-species variation, and important sire and dam differences. Sire generational effects, regardless of whether all offspring or only $G_{2+}$ offspring were examined, were consistent. This suggests that for a given species whatever effect captivity has on the survival of a male's offspring is maintained from the moment animals are brought in from the wild and continues through subsequent generations. For example, two species (red wolves and African wild dogs) showed strong positive effects of sire generation in both the full model and the $G_{2+}$ only model, while four species showed consistent negative effects. Compared to males, the results for females were more complicated. In the full model (incorporating all generations) the patterns of inter-species variation in the effect of generation were remarkably similar to the effects seen for sires (compare dam generation to sire generation, Fig. 2b). However, the effects of dam generation on $G_{2+}$ offspring were uniform and slightly negative across all the taxa we studied, even for those species that had strong positive responses when offspring of wild-born parents were included (Fig. 2b). This result suggests that regardless of whether captivity negatively impacts female breeders in the first generation or not, females breeding at generations $G_{2+}$ are likely to see slight declines in offspring survival across all our species studied.

Why do we see differences in offspring survival when comparing first to subsequent generations for females, but not males? Fitness changes over captive generations may occur as a result of genetic processes (both neutral e.g. drift[33], and non-neutral e.g. adaptive changes in allele frequencies[7]), non-genetic processes (e.g. behavioural change[34]), and/or epigenetic effects (e.g. maternal effects, transgenerational change[14,35]). These processes may act on the first generation, and/or across multiple generations, and could have different effects on males and females, for example due to differences in the provision of parental care[36]. Our use of a long-term measure of offspring survival to age of reproductive maturity (ranging from <1 year to >15 years across

the species in our dataset, Table 1), provides ample opportunity for a range of parental effects to influence survival[37]. Parental (transgenerational) effects can be investigated by performing experimental crosses of wild and captive parents, as seen in aquaculture settings[11,14]. However, in a conservation context involving highly threatened species, it is usually impractical to implement experimental breeding designs.

We saw important differences between the sexes when removing first-generation effects, which may be a result of the strength of maternal effects relative to paternal effects. In mammals (12 of the 15 species in our dataset), reproduction is generally a greater investment for females than males[36,38], so there may be greater opportunity for maternal effects to influence offspring survival than paternal effects. In our analysis, the birth origin of the dam (wild or captive) had a greater influence on offspring survival than that of the sire (Fig. 2b), a pattern that has been seen for other measures of reproductive success in captive populations[28,39,40]. An understanding of the life-history and parental investment of a species will assist in interpreting and addressing adaptive fitness changes.

Given widely reported negative effects of inbreeding on naturally outbreeding species and species managed in captivity[17,41,42], it is perhaps unsurprising that the strongest driver of an offspring's survival to reproductive maturity was its inbreeding coefficient (equivalent to the kinship of the parents). This trend was consistent across all species in our study. Importantly, our study is the first to our knowledge in a conservation setting to disentangle inbreeding effects from generations in captivity and time. Our dataset includes genetically depauperate species that have undergone substantial historic and/or recent bottlenecks such as the cheetah[43], Tasmanian devil[44] and red wolf[45]. Even these historically inbred species experienced inbreeding depression in captivity (Fig. 2b), suggesting that any purging to reduce the frequency of deleterious alleles is weak or ongoing. Minimising inbreeding is standard practice for captive management programs[19] including the ones analysed here. A growing interest in using group housing to promote more natural social settings and minimise adaptation to captivity[10] means that it may become harder to avoid inbreeding as genetically suboptimal pairings require additional management to prevent or detect in group settings.

Pedigree-based inbreeding coefficients such as those used in this study may underestimate true inbreeding as a consequence of the assumption in captive management that wild-sourced founders are unrelated. Molecular genetic methods have demonstrated that the assumption of unrelated founders does not always hold true[46,47], so molecular inbreeding estimates may improve modelling of survival in captivity. Genomic approaches tend to be species-specific (e.g. species-linked microsatellite markers, SNP arrays, whole genome resequencing) and so can provide inference on finer-scale diversity and relatedness patterns within a species[48]. Nevertheless, our pedigree-based results provide strong support for the recommendations of Leberg and Firmin[49] to avoid inbreeding in captive breeding programs, as purging is unpredictable and any benefits are unlikely to outweigh the costs of inbreeding depression[17].

Parental age at breeding had a large effect on offspring survival. We hypothesise that the positive effect of sire age (Table 2) may be a result of social factors[50], and age correlations with increased size[51] and experience[52]. On the other hand, we found younger mothers had higher offspring survival across the taxa studied, noting that some species showed exceptions when random slopes were fitted (Fig. 2b). This may seem unexpected given that the same factors linked to reproductive success in males above are likely to apply to females[53]; however, female reproductive senescence may swamp these effects through a reduction in offspring

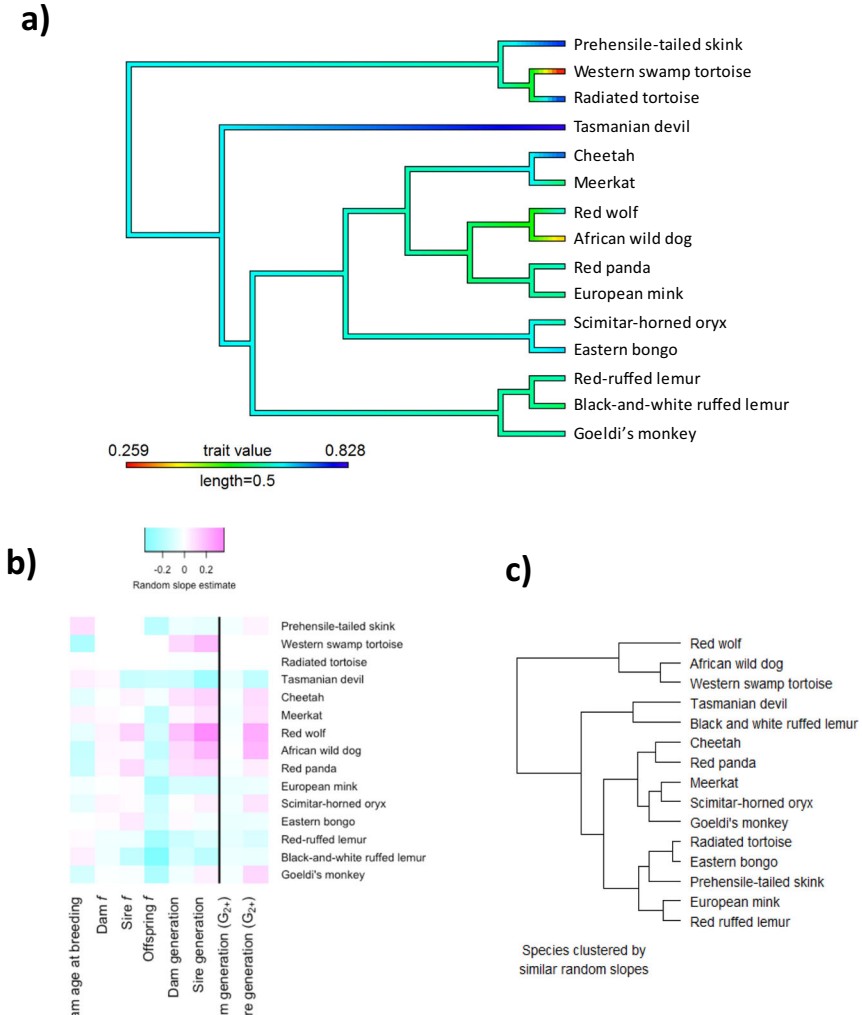

**Fig. 2 Phylogeny and random slope results. a** Phylogenetic relationships among the 15 species included in this study, shaded by the species' mean offspring survival (red = lower mean survival relative to other species in dataset, blue = higher mean survival). No evidence of phylogenetic signal was detected using this tree. **b** Heatmap of random slope estimates (i.e. magnitude of species-level effects of predictors on survival) for each parameter across the 15 species (*f* = inbreeding coefficient, pink = positive relationship between parameter and offspring survival, blue = negative relationship). Note that no random slopes could be estimated for sire age at breeding. The vertical black line separates the model with all offspring from the model with $G_{2+}$ offspring (no wild-born parents) only. **c** Dendrogram of species clustered based on similar random slope values, not phylogenetic relationships.

viability[54]. Our study did not investigate the causes of offspring mortality, nor did we investigate effects of parity on offspring survival[55], so testing of such biological hypotheses and interactions will require a more detailed species-level investigation.

This study was possible because studbook records are routinely collected in good captive management, enabling population managers to minimise inbreeding and maximise retention of wild-sourced genetic diversity[47]. We have therefore investigated generational change in conservation breeding programs without conducting experimental studies. The phylogenetic relationships among the species included in our dataset were not reflected in the species' offspring survival in captivity (clustered dendrogram of Fig. 2c does not correlate with phylogeny in Fig. 2a). Of the parameters we tested, only dam *f* showed evidence of statistically significant phylogenetic signal (Supplementary Table 1). It is possible that this result was driven by the strong negative impact of dam *f* shared by the two *Varecia* primates, and positive responses by more closely related canids (Supplementary Fig. 1). However, our study was not designed specifically to test

hypotheses around patterns in inbreeding across taxa, so we interpret the result with caution. Although comparative methods are useful in many fields of enquiry, such as assessing extinction risk[56], responses to climate change[57] and investigating captive welfare and stereotypic behaviours[58], in our case they were limited in their ability to predict species' responses to generations of captive breeding. Instead, larger studies containing more diverse taxa (e.g. a comparison of primates and canids) specifically designed to test such hypotheses, or species-level investigations, are necessary.

We have identified generational changes in fitness within captivity in diverse conservation breeding programs that are managed to avoid such a change. Further research is needed to investigate the possible underlying mechanisms of this change at a species level. We acknowledge that population-level genetic change in captivity is not intrinsically detrimental for individual animals being held in zoos, but may have potential negative consequences for reintroduction programs[59,60]. The results of this study demonstrate that generational changes in fitness are difficult to

predict, but are occurring in some long-running conservation breeding programs even with best-practice management.

## Methods

**Studbook data.** We obtained the international or regional studbooks of 15 species from the relevant regional zoo associations and studbook keepers, totalling 58,611 individuals, including 11 eutherian mammal, 1 marsupial and 3 reptile species (Table 1). These studbooks were selected on the basis of availability, size, taxonomic diversity, generations of captive breeding, and limited unknown ancestry. Species that are managed under a group rather than individual basis, such as the Rodrigues flying fox (*Pteropus rodricensis*), could not be included due to the large amount of unknown parentage. Birds were not included in this analysis as no studbooks were made available to us. Management practices specific to some bird species, such as double-clutching (also known as replacement clutching or egg harvest)[61], could bias survival estimates if not considered, yet are not typically recorded in studbook data. The pedigree management software PMx[62,63] was used to generate a dataset with one data point per offspring, containing information on the sire, dam, birth date, birth location, death date (if dead) and pedigree inbreeding coefficient ($f$). Pedigree-based inbreeding is calculated using known relationships and by assuming that wild-born founders are unrelated, so founders are assigned an inbreeding coefficient of 0, even though this assumption is not always met[46,64]. Individuals with unknown parents could not be included in the analysis; individuals with unknown ancestry further back in the pedigree could be included (using the PMx option "set unknown parents to wild"). We conducted further data cleaning in R[65] (version 3.5.1 – 4.0.1), whereby for each offspring we used the known parents to calculate age at birth, generations in captivity ($G_X$) and inbreeding coefficient ($f$) of the sire and dam. Wild-born animals are assigned generation $G_0$. For captive-born animals, generation is calculated as the average generation of the parents plus 1, meaning that it can be a non-integer e.g. $(G_0 + G_1)/2 + 1 = G_{1.5}$. We also calculated the age at death of the individual, or current age if still alive. We truncated the last 364 days of data from the studbook to minimise the possibility that recent deaths had not yet been updated in the studbook.

For each species, we defined the age at reproductive maturity for each sex in days using the AnAge (Animal Ageing and Longevity) database[66], or PMx for the species without data in AnAge (all three reptile species, red wolf, and male scimitar-horned oryx). We prioritised AnAge values where possible, due to potential biases in the age of reproduction in captivity (e.g. delaying reproduction to minimise genetic adaptation to captivity[10]). We excluded all individuals born within the timeframe of the reproductive maturity age from the 364 days before the current date of the studbook, as these animals would not yet have had the opportunity to reach reproductive maturity. We removed individuals that had been identified as hybrids in the red wolf studbook, and those that were born in the wild or released to the wild before the age of reproductive maturity (affected the red wolf and Tasmanian devil studbooks). A further 13,089 individuals did not have a known sire or dam or both, meaning that predictors of interest were unknown, and these individuals were excluded. Multiple imputation could not be attempted as, for 8530 individuals, all values of interest were missing. Of the remaining 37,493 individuals with complete data, we established whether they had survived to the defined age of reproductive maturity (1) or not (0). For individuals with unknown sex, we defined age at reproductive maturity as the shortest of the two sexes. Data appeared to be missing at random with respect to time.

There was high variation in our predictors of interest between species (Table 1), due to historic captive management and variation in species biology. We therefore standardised numeric variables of interest (dam/sire $f$, dam/sire generation, dam/sire age at breeding and offspring $f$) within each species by centring on the mean and dividing by 1 standard deviation with the 'standardize' package[67] to avoid species with extreme values unduly influencing the results and to assist interpretation of model parameter estimates. This method of standardising does not affect the relative variances around predictors.

**Phylogenetic correlations.** The 15 species (Table 1) varied in their mean offspring survival (as expected due to varying life-history traits e.g. R- vs. K-selected species, single vs. multiple offspring, differential maternal investment; and variation in population management, e.g. age at accession to studbook). We assessed phylogenetic correlations in our dataset by creating a tree in the 'rotl' package[68] based on phylogenies available through the Open Tree of Life[69]. The topology of the tree was used to calculate lambda, an estimate of phylogenetic signal ranging from 0 (no signal) to 1, and a likelihood ratio test used to determine statistical significance at $\alpha = 0.05$. Phylogenetic signal would indicate that closely related species are more similar in their offspring survival rates than distantly related species. As species varied in their mean offspring survival, but phylogenetic signal was very weak, we proceeded to model offspring survival controlling for variation among species, but not phylogenetic relationships among species.

**Random factors and model fitting.** Generalised linear mixed models (GLMMs) were fit in 'lme4'[70] with a binomial response and a nested random factor design. The random factors we controlled for were Species, Birth Program and Year. The Species random factor controls for variation among species in their life history (such as their generation length, i.e. how rapidly the species breeds) and captive management. Birth

Program refers to the region where an individual was born as defined by ZIMS (Africa, Australasia, East Asia, Europe, Latin America, Middle East, North America, South East Asia, South Asia, unknown and other). Individual zoo information was available for most species, but the distribution of records across zoos was highly uneven as some species were bred only at a select few specialist institutions, while other institutions had only a small number of breeding records over the entire study period, which would likely cause model convergence issues. Estimating survival at a particular zoo is further complicated by transfers of animals between zoos, and whether survival is influenced by conditions at birth, or later, or both. Pooling zoo information into Birth Program simplifies these challenges, by assuming that animals are more likely to be transferred within a region than to a different region, as the region is unlikely to change throughout an animal's early life to reproductive maturity. Birth Program was nested within Species to account for regional specialisation. For example, while a region may have particularly high offspring survival of one species, it may have below-average offspring survival of a different species. This can be due to a range of factors including taxonomic expertise, climate, population management practices and varied husbandry. The year of birth controls for improvements in offspring survival made over time with improved husbandry or other events such as intakes of wild animals after the establishment of the captive population, and was also nested within Species as the studbooks covered very different time-frames (the year of first captive-born offspring ranged from 1881 [scimitar-horned oryx] to 1991 [Western swamp tortoise]).

**Independent litter sampling.** Animals born as part of the same litter or clutch share the same dam and often the same sire. We identified litters as animals born to the same dam on the same day for mammalian species, and as animals born to the same dam in the same year for the tortoises and skink. One offspring from each litter was randomly selected ($N = 21,282$ independent individuals). Random selection of independent litter-mates was repeated a total of five times and analyses pooled to obtain model estimates. Our global model consisted of:

Survival ~ Dam generation + Sire generation + Dam age at breeding + Sire age at breeding + Dam $f$ + Sire $f$ + Offspring $f$ + (1|Species/Birth Program) + (1 | Species:Year)

We examined model fit using the 'DHARMa' package[71], by calculating Variance Inflation Factors (VIFs) to ensure multi-collinearity of predictors was <2[72] and satisfying the Kolmogorov–Smirnov test of uniformity, outlier test, non-parametric dispersion test and zero-inflation test.

**Model selection.** We conducted model inference under an information theoretic approach following Grueber et al.[73]. All possible sub-models were fitted using the 'dredge' function from the 'MuMIn' package[74], and models within the top 2 $AIC_C$ of the top model were retained and model averaged (conditional average method). We interpreted predictors based on the size, direction and precision of the model estimate and its relative importance (sum of Akaike weights for top models containing the predictor).

**First-generation vs. multi-generational changes.** We may expect differences in the survival rates of offspring of wild-born parents vs. the offspring of captive-born animals. Therefore, we ran a second analysis as above but excluding all offspring with either one or both wild-born parents (i.e. $G_{2+}$ offspring). We repeated the $G_{2+}$ analysis using the five independent litter sampling subsets ($N = 16,514–16,516$ offspring, sample size varies because litter-mates identified by the same dam may have a different sire, and the generations in captivity of the sire may vary).

**Sensitivity testing.** We re-ran the global model and model selection steps with an extended dataset of $N = 37,484$ individuals (nine data points with high leverage identified in residual plots removed). The extended dataset ignores relationships between litter-mates (it was not possible to fit litter/clutch-level random effects) so does not statistically account for biological non-independence. Three of the species in our dataset typically give birth to only one offspring and would be unlikely to drive any differences between the extended dataset relative to our main analysis (Supplementary Fig. 3). Results of the extended dataset model were used to qualitatively check that our random selection of independent litters did not unintentionally bias the dataset, by comparing parameter estimates between the two approaches. Qualitative inferences were the same as in our main analysis (Supplementary Table 2a, Fig. 1). We also repeated the $G_{2+}$ model with our extended dataset for sensitivity testing ($N = 27,734$ offspring) and found qualitatively similar parameter estimates (Supplementary Table 2b, Fig. 1), indicating our approach did not bias results.

**Random slope models for between-species variation.** We selected one of the independent litter sampling subsets that was representative of the five sets of results to further investigate trends across species (Supplementary Fig. 2). We fitted separate random-slope models for each predictor to estimate species-level effects, where the main effect is interpreted as the mean across all species, and the random component quantifies the amount of variation in that slope among species. For example, while the main effect may suggest a negative relationship between dam age at breeding and offspring survival over all species, the effect of dam age may vary between species. Random-slope models were fitted from the global model, as model averaging cannot provide random slope estimates. No parameters were dropped

from the top model set with model averaging (Supplementary Table 3), so we do not expect the random slopes models to differ substantially if estimation after model selection was possible. We also investigated phylogenetic signal by estimating lambda using the random slope estimates for each of the parameters of interest aside from sire age for which a random slope model did not converge. Additionally, we fitted random slopes model from the $G_{2+}$ independent litter sampling subset for the dam and sire generation parameters, as we may expect species responses to differ after removing the offspring of wild-born animals from the analysis.

**Reporting summary**. Further information on research design is available in the Nature Research Reporting Summary linked to this article.

## Data availability

Data underlying this analysis is available as Supplementary Data 1. All figures can be reproduced using this data and the available code. The Animal Ageing and Longevity (AnAge) Database is available at: https://genomics.senescence.info/species/.

## Code availability

Custom R code underlying this study is available as Supplementary Code 1. All figures can be recreated using this code.

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

## Acknowledgements

Our research into adaptation to captivity in conservation breeding programs is supported by the Australian Research Council Discovery Projects (grant DP170101253 to CEG), San Diego Zoo Global (including support for CEG's previous fellowship, during which a part of this work was undertaken), The University of Sydney and by an Australian Government Research Training Program (RTP) Scholarship and Murray Writing Up Support Grant to KAF. We are grateful to the regional zoo and aquarium associations for their participation in this study, including the Association of Zoos and Aquariums (AZA, North America), European Association of Zoos and Aquaria (EAZA), Zoo and Aquarium Association (ZAA, Australasia) and Pan-African Association of Zoos and Aquaria (PAAZA), and in particular Kristine Schad, John Werth, Chris Hibbard, James Biggs, Kelvin Limbrick, Sarah Long, Danny de Man, Candice Dorsey and Kristin Leus. We thank the studbook keepers and conservation managers for providing studbook data and management insights, including in alphabetical order Angela Glatston (Rotterdam Zoo), Antoinette Kotze (National Zoological Gardens of South Africa), Carla Srb (Zoos Victoria), Dani Jose (Perth Zoo), Kate Vannelli (Cheetah Conservation Fund), Katie Kimble (Toledo Zoo), Laurie Marker (Cheetah Conservation Fund), Lydia Bosley, Mark Warneke (Chicago Zoological Society), Michael Ogle (Zoo Knoxville), Mylisa Whipple (Saint Louis Zoo), Sam Curtis (Sacramento Zoo), Tania Gilbert (Marwell Wildlife), Tiit Maran (Tallin Zoo), Tracy Rehse (National Zoological Gardens of South Africa) and Will Waddell (Point Defiance Zoo and Aquarium). Data visualisation was improved with the helpful suggestions of Charles Foster and other members of the Evolutionary and Applied Zoology Group (University of Sydney). We thank M. Christie, A. Charmantier, W.B. Sherwin for constructive feedback on an earlier version of this manuscript.

## Author contributions

K.A.F. obtained studbook data, performed data cleaning and analysis, prepared figures and tables and drafted the manuscript. C.J.H. assisted in obtaining and converting studbook data, provided conceptual guidance on the analysis, critically revised the manuscript and oversaw the project. C.E.G. provided technical advice on data analysis, critically revised the manuscript and oversaw the project.

## Competing interests

The authors declare no competing interests.
