## [Peer Review File · Nature Communications]

RESPONSE TO REVIEWER COMMENTS

NCOMMS-20-27956

Reviewer #1 (Remarks to the Author):

In this manuscript, Farquharson et al. present an ambitious study using captive breeding data studbooks from 15 species to quantify the evolution of fitness in captivity. Because of our growing needs for captive breeding, these data and analyses are timely and relevant for on-going conservation and management actions. Overall, I was impressed with the breadth of analyses completed and have only minor and clarity suggestions.

I find it incredibly difficult to understand the analyses of this manuscript when reading from top to bottom. For example, Line 85 starts the beginning of the results section talking about negative slopes presumably from a regression, although at this point in the manuscript the contents of the regression are unclear. The text directs to figure 1b, which is talking about random slopes, table 2, which is talking about an as-yet undefined random selection procedure, and Figure 2, a forest plot. This issue repeats in later Results-section paragraphs. This could be fixed by providing some explanations of what was done alongside the results.

Response: We have added text at the beginning of the Results section to give context to our overall approach including our regression modelling without reiterating the methods that are described in detail later in the paper. Importantly, we have clarified our two main approaches as the “main (overall) effects” and “species-level effects” (L103 – L118). The results have been restructured for flow and we have clarified references to the figures/tables so that they match with the approach discussed. As a result, the order of some figures and tables has now changed.

Line 83: The beginning of the results seems like a great place to give some information about the pedigree data you used. For example, on average how many generations of data were included in the pedigrees?

Response: We have added a new section at the beginning of the Results “Pedigree information provides a rich source of data for retrospective analyses of captive breeding trends” (L85 – L102). This section now includes some basic information about the pedigrees including the number of generations of data, supported by Table 1.

Line 252-254: I am guessing you used PMx to estimate age at which species reproduced to get an estimate of age at reproductive maturity, although this is not exactly clear. If this is the case, why not use PMx for all of your species? It seems like it would be preferable to use a known age rather than a species-wide estimate?

Response: Captive breeding management may bias the average age at first reproduction, relative to the natural age of reproductive maturity in the wild. For example, one strategy to minimise adaptation to captivity is to delay the age at first breeding (Williams & Hoffman; 2009). Furthermore, taking the earliest age of reproduction recorded in the studbook could be influenced by errors or

outliers in captivity that do not represent the typical age of reproductive maturity. Therefore, we used the AnAge database (which is independent of the studbook data), where records existed, as now noted in the Methods (L316 – 318).

Williams, S.E. & Hoffman, A. (2009). Minimizing genetic adaptation in captive breeding programs: a review. *Biological Conservation*, **142**(11): 2388-2400.

Line 284: Was inbreeding coefficient related to phylogeny? Asked another way, did F increase unevenly across species?

Response: We previously modelled phylogenetic signal of survival to reproductive maturity. Now, we have added estimates of phylogenetic signal for each of the parameters we investigated (offspring f , dam f , sire f , dam age, dam generation and sire generation; with the exception of sire age for which a random slope model did not converge). To do this, we extracted the random slope estimates (species-level effects) and calculated lambda (measure of phylogenetic signal) (L429 – 432). The results are in Supplementary Table 1.

Offspring inbreeding coefficient (which had the strongest relationship with offspring survival) did not show evidence of phylogenetic signal, however dam f had a statistically significant lambda estimate (P-value from likelihood ratio test = 0.0258), suggesting phylogenetic signal in the effect of dam f on survival. This result was likely driven by a strong negative relationship observed for the two *Varecia* primates. To illustrate this result, we now provide a phylogeny of our study species shaded by the estimate of dam f on survival as Supplementary Figure 1. No other parameters showed evidence of phylogenetic signal. Our data were not collected with the specific aim of testing hypotheses around inter-species variation in responses, so we recommend that datasets containing a larger number of more diverse taxa are used to test this further. We have added this recommendation to our Discussion (L266 - 277).

Figure 2. It is not clear to me why model estimate is on the x axis. Shouldn't the variables be on the independent axis?

Response: Figure 1 (formerly Figure 2) is a forest plot of effect sizes, a common graph used to summarise information from multiple sources, which originated in meta-analysis. As this figure summarises information from four models across seven parameters, we believe it is the clearest way to visualise and compare the magnitude of the effects of our study variables. We have added the explanation “positive estimates represent a positive relationship with offspring survival to reproductive maturity” to the figure caption to improve interpretation.

Reviewer #2 (Remarks to the Author):

Authors here provide an extensive summary on offspring survival in captive breeding studying an enormous data collection based on 15 species. While some species have been partly looked at such a comprehensive summary is surely an important contribution for conservation. Apparently fitness change effects across time and generations can be expected, which cannot easily be explained by known mechanisms (such as inbreeding depression). Authors follow up on a previous study investigating birth origin effects on reproductive success and take their previous analyses a step forward for longer time periods. Overall this is a clear written manuscript and the methods are

exactly described and authors have been cleaning their data rigorously and controlling for potential errors. I have only a few minor comments here:

1) How have species been selected? That is only briefly managed and I assume that is based on data availability. However, it would be interesting to see what thresholds have been used here – so how many species needed to be excluded? Obviously mammals are a focus, but it is surprising to see that there are no birds included. Please add a bit of information here.

Response: Studbook data was obtained by reaching out to regional zoo associations (Australasia, USA, and Europe) for data, and so only studbooks kindly shared with us could be included. Furthermore, to test our hypotheses, we were restricted to only datasets that had clear parentage records. We have now added more detail of the types of studbooks that were not included to the Methods: “Species that are managed under a group rather than individual basis, such as the Rodrigues flying fox (*Pteropus rodricensis*), could not be included due to the large amount of unknown parentage. Birds were not included in this analysis as no studbooks were made available to us. Management practices specific to some bird species, such as double-clutching (also known as replacement clutching or egg harvest)⁶¹, could bias survival estimates if not considered yet are not always recorded in studbook data” (L292 - 298).

2) The “random slope” in graphs is not easy to understand – please do provide an explanation of what that means, so any interested reader can get an idea of what this ecologically responds to

Response: As above, we now use the term “species-level effects” throughout to clarify the purpose of the random slope analyses. “Random-slope estimates from GLMMs were obtained to quantify variation in the responses of each of the 15 species to captive breeding. As with our main effects, a positive random slope estimate represents a positive relationship with offspring survival for that species” (L113-118).

3) I have been a bit confused about the generation time. See comments below. You measure “Generation” but how is that related to “generation time” – please add an explanation here.

Response: “Generation time” typically refers to the generation length, i.e. the average age of the parents of a population, and reflects the breeding rate of a species. Our study only investigates captive generations, i.e. the number of generations of captive breeding an individual has experienced, and not generation time/length specifically. We have now added a paragraph to the beginning of the results explaining how generations in captivity was calculated and how it varies across our dataset (L93 – 102). There is a relationship between captive generations and generation time/length, because over a given time period a fast-breeding species will experience more generations of captive breeding than a slow-breeding species. By accounting for variation among species in our models (“Species” fitted as a random factor), we minimise the confounding impact of generation length on our results. This is now clarified in the Methods (L354 - 356).

4) I also think a bit more explanation on the calculation of N_e is needed here. It is just referred to as the programme output, but I would like to see a bit more information here, without needing to go to the manual.

Response: We obtained effective population size for each species from the studbook data in PMx, reported in Table 1. PMx calculates N_e as defined by Wright (1931).

Jimenez-Mena, B., Schad, K., Hanna, N. & Lacy, R.C. (2016). Pedigree analysis for the genetic management of group-living species. *Ecology and Evolution*, 6(10): 3067-3078.

Wright, S. (1931). Evolution in Mendelian populations. *Genetics*, 16: 97-159.

5) What I did not get from the methods section: Have there been for some species multiple times where animals from the wild have been added. I can understand the treatment of data, but I wonder how that has been accounted for

Response: Yes, sourcing wild animals over multiple years does occur in some captive breeding programs where the opportunity still exists. Wild animals are assigned generation 0 and their offspring generation 1, regardless of the point in time that they were brought into captivity. We controlled for variation in the year of birth by fitting Year as a random factor, nested within Species. If for example, there was a large influx of wild animals of a species in the year 2000 and their offspring in 2001 had particularly high survival, the variance between year of birth would be accounted for in the random component of the model so that the fixed component is estimated independently (i.e. our generation parameters are estimated after accounting for variance between years for that species). We have expanded: “The year of birth controls for improvements in offspring survival made over time with improved husbandry or other events such as intakes of wild animals after the establishment of the captive population, and was also nested within Species as the studbooks covered very different time-frames (the year of first captive-born offspring ranged from 1881 [scimitar-horned oryx] to 1991 [Western swamp tortoise])” (L371 – 376).

6) Getting the studbook information is a massive effort – however, I guess that some species have been in zoos at different location and data have been pooled. Did you actually account for the location/origin of your species in the model? Can you find effects independently of the location? Obviously if these are pooled samples that is also effecting population size. I am sure you have thought of that, but please add an explanation here.

Response: Hundreds of zoos were involved across the entire study, however the distribution was very uneven across the dataset so attempts to fit precise “location” would impact model convergence (i.e. some species are bred at a select few institutions; and the amount of data across zoos varies considerably over time: our dataset represents more than a century of records). Additional complexity arises considering that animals are commonly transferred between zoos – should the offspring location at birth, or death, contribute the most to offspring survival? We were therefore unable to model differences between institutions, however we instead pooled data within regions (“Birth Program”), under the assumption that few animals move between regions from their region of birth. Regional differences were included in the models as a random effect, nested within Species to control for large-scale differences. We have included this additional information at L358 – 367.

12: vertebrate species could go into the text here

Response: Added (L12).

18: what is a evolutionary relationship? Please define

Response: We meant the phylogenetic relationships among species, i.e. how evolutionarily close or distant the species were from each other. Now reads: "...responses to captivity could not be predicted from species' evolutionary (phylogenetic) relationships" (L19) and further described in Results (L111) and Methods (L344).

20: so what is the conclusion here – I feel that a final concluding remark or discussion is needed for the abstract – at the moment you leave it with the pure statement.

Response: Our final statement reflects the main and novel finding of our study: "Even under best-practice captive management, generational fitness changes that cannot be explained by known processes, such as inbreeding depression, are occurring." (L19). We did not investigate what impact fitness changes in captivity may have once animals are reintroduced to the wild so do not want to introduce speculation.

49: Other – what specific do you refer to? you only generally mention heritable effects earlier

Response: At L46 we mention non-genetic effects such as husbandry, nutrition, or maternal effects (which may be heritable or non-heritable). We hypothesise that non-genetic effects may result in short-term first-generation fitness changes that may differ from longer-term multi-generational changes that may indicate heritable changes.

60: the "must" is a pretty strong statement –and I tend to disagree here

Response: Our statement that the diversity of species bred in captivity must increase to prevent extinctions is supported by the commentary of Fa et al. (2014) (reference 21), which draws upon other studies such as Martin et al. (2014) (reference added), that the diversity of species held by zoos does not reflect the diversity of threatened species globally. There are various reasons for this and our statement is not intended as a criticism of the role of zoos in conservation, rather as a challenge that still remains. We have modified our statement: "Globally, the diversity of species bred in captivity must increase if extinctions are to be prevented²¹: while a great variety of species are bred in zoos, this variety does not reflect that of the diversity of threatened species for a range of reasons, which may be challenging to overcome²²" (L62).

83: why "despite" – is that something so contrary or wouldn't you expect changes in survival anyway?

Response: Despite was the wrong word choice. We have modified this to "Offspring survival changes over generations of captive breeding, independently of substantial inbreeding depression" to reflect our findings (L119).

247: I don't understand what generation here means - I guess these are the values in the table, but how do those differ to generation time? I would have expected some values on generation time here.

Response: See response to comment 3 above: we now clarify what we mean by generations in captivity and how it was calculated earlier in the manuscript (L93 – 102).

Table1: what is generation? I guess it is time? and how does that fit ? if it is years ? then it does not fit to the age or breeding in days – I am a bit confused on that.

Response: Edited Table 1 header to specify “Captive generations” (the number of generations of captive breeding experienced), as distinct from generation length/time. Captive generations are not measured in time, for example a wild animal is assigned generation 0 and its offspring generation 1 regardless of how quickly the offspring was produced. As above, we have clarified the meaning of captive generations in L93 - 102.

Figure 1: so what does a random slope mean then? You need to explain that to the broader readership.

Response: See above (L113 – 118). The caption for Figure 2 (formerly Figure 1) now includes: “Heatmap of random slope estimates (i.e. magnitude of species-level effects of predictors on survival)”.

Reviewer #3 (Remarks to the Author):

Manuscript: Offspring survival changes over generations of captive breeding: insights from 15 threatened species

Katherine A. Farquharson, Carolyn J. Hogg & Catherine E. Grueber

Farquharson et al. conducted an extensive study analyzing existing studbook data from 15 species to investigate factors influencing offspring’s survival to age of reproductive maturity in captivity. They used linear mixed models to test the effect of variables such as dam and sire inbreeding, offspring inbreeding, dam and sire generation, and dam and sire age at breeding, by controlling for year effects, regions and species taxonomy. Their study includes over 37,484 individuals represented in studbooks from 11 eutherian mammals, 1 marsupial and 3 reptile species. Main findings of their study include that survival effects in captivity cannot be predicted by taxonomic relationships or human management, so species-level investigations are necessary, and the strongest driver of offspring’s survival to reproductive maturity is individual’s inbreeding coefficient.

The linear mixed models used by authors to test the effect of pedigree variables in offspring’s survival are appropriated, and considerations about first- versus multi-generational analyses were informative to separate effects coming from breeding wild parents versus captive parents. Additionally, probabilistic results across models tested were consistent when sampling was randomized by considering one individual per litter/clutch, supporting the robustness of the methodological approach implemented.

This manuscript is well written and it addresses long-standing questions among conservation and population biologists, and managers about predicting long-term viability and sustainability of captive populations (via offspring survival) using comparative approaches, and identifying pedigree variables relevant for managing captive populations. I recommend this manuscript for publication in Nature

Communications because of the extensive comparative approach and use of a phylogenetic framework to predict offspring's survival among species. However, it is fair to say that the effect of individual's inbreeding coefficient in offspring's survival to reproductive maturity documented in this study is not a new idea, and it has been observed before in other species (e.g. Hawaiian crows, Hoeck P.E. et al., 2015. *Biol Conserv*, 184:357–364).

Response: Thank you for your feedback. We agree that the impact of inbreeding on survival has been well-documented and think that our work adds to that literature base. We have now added additional citations to the Discussion (L226). Given that inbreeding has such a strong impact on survival, we thought it was important to include it in our models so that it would be accounted for when examining other factors of interest such as the number of generations in captivity. Importantly, our study is the first in a conservation setting to comprehensively disentangle inbreeding effects from generations in captivity and time.

Comments and suggestions:

- The first comment is about assumptions made in the analyses regarding wild-born founders as unrelated individuals, so their inbreeding coefficient is 0. I understand the difficulty in validating such assumptions but I think it is important to highlight the potential effect of related founders in overall estimations of inbreeding coefficients across the pedigree, particularly in the first generations, as indicated in several publications by Ivy J. and Lacy R.C. (e.g. Ivy J. et al., 2016. *Journal of Heredity* 107), and as consequence, effects in offspring's survival. Captive populations are in general managed using mean kinship estimates and these values can be underestimated if founders are related. I suggest to expand this topic in the discussion of the manuscript as potential confounding factor, and highlight the benefits of using genetic/genomic approaches to validate pedigree data.

Response: Our use of pedigree data does not allow us to assess molecular inbreeding, but it is true that pedigree-based inbreeding coefficients can be underestimated if founders are related. We have added the assumption of unrelated founders and implications of this assumption to the Discussion (L238 - 245) along with relevant citations.

- The phylogenetic signature of offspring's survival among species was determined considering specie's mean offspring survival. I wonder whether describing survival trends in terms of variables tested in the models is more appropriate. This means, for example, to describe whether all carnivores studied are similarly affected by sire or dam generation in terms of offspring's survival, or any other trend observed between artiodactyl species, or among primates and reptiles. This approach may provide evidence of relevant variants to be considered when managing certain populations in captivity within groups of vertebrates, beyond consideration of inbreeding coefficient.

Response: This was also touched on in our response to Reviewer 1. We have now calculated phylogenetic signal across the dataset for each of our parameters of interest, using the random slope estimates for each species within each parameter. This is now described in Methods (L429) and Results (L125) with Supplementary Table 1 displaying the lambda estimates for phylogenetic

signal and associated significance tests. Only *Dama f* showed evidence of statistically significant phylogenetic signal (illustrated at Supplementary Figure 1). However, our dataset of 15 species may be too small to rigorously examine hypotheses that were not central to our aim, such as trends among particular taxa (e.g. we have only 3 reptiles and 3 primates in the dataset). We have added a recommendation for future studies designed specifically to test such hypotheses (L275).

- This study has tested with linear mixed models, effects coming from pedigree and studbook variables, but I wonder how important is to also add information regarding species' level of endangerment (e.g. IUCN categories) or historical demographic trends that might shed lights on the genetic make-up of species and the real consequences of inbreeding in the survival of populations. Recently, genomic data have been generated for a large number of species of conservation concern that are also managed in captivity, and the use of these data has allowed estimating the evolutionary and demographic histories of species. This information has served to understand genome-wide levels of inbreeding (e.g. measured as runs of homozygosity) and genetic load, as factor contributing to inbreeding depression. By knowing the status of the gene pool of species at the moment of establishing captive breeding programs (represented by founders), and controlling for factors like genome-wide inbreeding and genetic load in predicting models, I wonder whether the predictive power of offspring's survival can improve by eliminating confounding factors caused by different demographic and evolutionary histories, even between closely related species. I understand the complexity in creating such models but genetic and historical demographic factors, in addition to life history traits and management practices, are the variables where the real power is in terms of predicting sustainability of wild and captivity populations. Examples include the work recently conducted in *Dama gazelle* and black-footed ferret by Klaus Koepfli at the Smithsonian Conservation Biology Institute. It might be of interest to highlight in the discussion of the manuscript future directions in this field aiming to combine traditional management strategies using pedigree data with genomic information.

Response: We have now added the IUCN Red List status of each of the species to Table 1. We agree it would be interesting to assess captive breeding survival against endangerment or other parameters, however don't think that it is appropriate to do so with our dataset. Our dataset was not collected with these aims in mind and is not representative of Red List trends. For example, of our 15 species, 13 are threatened with extinction (the meerkat is Least Concern and the prehensile-tailed skink is not listed but is a CITES Appendix II species). This compares to the global assessment of 27% of all assessed species listed as threatened. We believe there is not enough variation in threat status within our 15 study species to avoid spurious correlations in trends with survival. Instead, we now recommend studies designed specifically to test such hypotheses as a future direction (L275).

We agree with the utility of genomic data for assessing survival trends at a finer scale. Our study made use of available pedigree data, and demonstrates the utility of a retrospective approach where data is already available and reasonably standardised across biologically diverse taxa. Although genomic data can be extremely informative, we did not have such data available to this project. Nonetheless we agree this is an important future direction and have noted this in the discussion (L239 - 245).

References mentioned:

Jamie A Ivy, Andrea S Putnam, Asako Y Navarro, Jessica Gurr, Oliver A Ryder, Applying SNP-Derived Molecular Coancestry Estimates to Captive Breeding Programs, *Journal of Heredity*, Volume 107, Issue 5, September 2016, Pages 403–412, <https://doi.org/10.1093/jhered/esw029>

Willoughby, J. R., Ivy, J. A., Lacy, R. C., Doyle, J. M., & DeWoody, J. A. (2017). Inbreeding and selection shape genomic diversity in captive populations: Implications for the conservation of endangered species. *PLoS ONE*, 12(4)

Hoeck PEA, Wolak ME, Switzer RA, Kuehler CM, Lieberman AA (2015) Effects of inbreeding and parental incubation on captive breeding success in Hawaiian crows. *Biol Conserv* 184:357–364

NCOMMS-20-27956A

Reviewer #1 (Remarks to the Author):

I remain impressed with the breadth of analyses completed by Farquharson et al. to quantify the evolution of fitness in captivity. I am happy with the response/revisions based on my previous review and have found only a few typos that should be corrected prior to publication.

Line 109: The "and vice versa" clause is not strictly true at least how the regressions were run; please remove.

Response: Now reads "A positive regression estimate indicated a positive effect of the predictor on offspring survival, conversely, a negative regression estimate indicated a negative relationship with offspring survival." (L111).

Line 110: 'is interpreted' should be 'was interpreted'

Response: Corrected (L113).

Line 161: Should species taxonomy be species phylogeny?

Response: Corrected (L162).

Reviewer #2 (Remarks to the Author):

Authors have thoroughly addressed all reviewers comments and I think the issues mentioned have been solved now.

I have only a few, more stylistic, comments where I found the manuscript difficult to understand.

11 : why "but" isn't it "and" ?

Response: Corrected (L10).

20-21: I found the end of the sentence confusing.

Response: Now reads “Even under best-practice captive management, generational fitness changes that cannot be explained by known processes (such as inbreeding depression), are occurring.” (L18-19).

227.: something missing here? or “that” wrongly?

Response: Clarified, “... it is perhaps unsurprising that the strongest driver of an offspring’s survival to reproductive maturity was its inbreeding coefficient...” (L226).

237: not sure I fully understand the sentence here. Do you argue that for some species (as an extreme the big apes) are more prone to inbreeding if managed in groups – and what would the consequence be?

Response: We argue that housing captive animals in groups (a suggested strategy to minimise adaptation to captivity) may make it harder to avoid inbreeding. This argument is not specific to any particular taxa. Without additional reproductive management such as the use of contraceptives, group-housing animals risks genetically suboptimal pairings that may not be detected unless molecular parentage analysis is undertaken. Revised to “A growing interest in using group housing to promote more natural social settings and minimise adaptation to captivity means that it may be harder to avoid inbreeding as genetically suboptimal pairings require additional management to prevent or detect in group settings.” (L236).

Reviewer #3 (Remarks to the Author):

Nature Communications - Review

Offspring survival changes over generations of captive breeding: insights from 15 threatened species

Manuscript #: NCOMMS-20-27956A

The manuscript resubmitted by Farquharson et al. entitled “Offspring survival changes over generations of captive breeding: insights from 15 threatened species (NCOMMS-20-27956A)” was carefully revised to address concerns and comments from reviewers in relation to analyses and clarifications of methods and results. For instance, additional text was added at the beginning of the Results section clarifying the approaches used to examine drivers of offspring survival in captivity (description of main effects and species level effects). Also, a new paragraph was included in the Discussion section clarifying the potential underestimation of pedigree-based inbreeding coefficients as a consequence of assumptions considering wild-sourced founders unrelated. I appreciated the effort of the authors in addressing the question related to the phylogenetic signature of offspring’s survival among species determined by mean offspring survival. The authors added in the revised manuscript new analyses estimating the phylogenetic signal across the dataset for each of the parameters of interest, using the random slope estimates for each species within each parameter. Results from this analysis are presented in Supplementary Table 1.

Overall, I consider the revised version of the manuscript highly improved and I support its publication, in the current format, in Nature Communications.

Response: Thank you.